# Risk Factors for Malnutrition among IBD Patients

**DOI:** 10.3390/nu13114098

**Published:** 2021-11-16

**Authors:** Larisa Einav, Ayal Hirsch, Yulia Ron, Nathaniel Aviv Cohen, Sigalit Lahav, Jasmine Kornblum, Ronit Anbar, Nitsan Maharshak, Naomi Fliss-Isakov

**Affiliations:** 1Sackler School of Medicine, Tel Aviv University, Tel Aviv 6997801, Israel; ayalh@tlvmc.gov.il (A.H.); yuliar@tlvmc.gov.il (Y.R.); nathanacohen@gmail.com (N.A.C.); jasminek@tlvmc.gov.il (J.K.); ronitan@tlvmc.gov.il (R.A.); nitsanm@tlvmc.gov.il (N.M.); naomifl@tlvmc.gov.il (N.F.-I.); 2Department of Gastroenterology and Liver Diseases, Tel Aviv Medical Center, Tel Aviv 6423906, Israel; Sigalitla@tlvmc.gov.il; 3Nutrition and Dietetics Department, Tel Aviv Medical Center, Tel Aviv 6423906, Israel

**Keywords:** malnutrition, screening, inflammatory bowel disease

## Abstract

(1) Background: Malnutrition is a highly prevalent complication in patients with inflammatory bowel diseases (IBD). It is strongly associated with poor clinical outcomes and quality of life. Screening for malnutrition risk is recommended routinely; however, current malnutrition screening tools do not incorporate IBD specific characteristics and may be less adequate for screening these patients. Therefore, we aimed to identify IBD-related risk factors for development of malnutrition. (2) Methods: A retrospective case-control study among IBD patients attending the IBD clinic of the Tel-Aviv Medical Center for ≥2 consecutive physician consultations per year during 2017–2020. Cases who had normal nutritional status and developed malnutrition between visits were compared to matched controls who maintained normal nutritional status. Detailed information was gathered from medical files, including: demographics, disease phenotype, characteristics and activity, diet altering symptoms and comorbidities, medical and surgical history, annual healthcare utility, nutritional intake and the Malnutrition Universal Screening Tool (MUST) score. Univariate and multivariate analyses were used to identify malnutrition risk factors. The independent risk factors identified were summed up to calculate the IBD malnutrition risk score (IBD-MR). (3) Results: Data of 1596 IBD patients met the initial criteria for the study. Of these, 59 patients developed malnutrition and were defined as cases (*n* = 59) and matched to controls (*n* = 59). The interval between the physician consultations was 6.2 ± 3.0 months, during which cases lost 5.3 ± 2.3 kg of body weight and controls gained 0.2 ± 2.3 kg (*p* < 0.001). Cases and controls did not differ in demographics, disease duration, disease phenotype or medical history. Independent IBD-related malnutrition risk factors were: 18.5 ≤ BMI ≤ 22 kg/m^2^ (OR = 4.71, 95%CI 1.13–19.54), high annual healthcare utility (OR = 5.67, 95%CI 1.02–31.30) and endoscopic disease activity (OR = 5.49, 95%CI 1.28–23.56). The IBD-MR was positively associated with malnutrition development independently of the MUST score (OR = 7.39, 95%CI 2.60–20.94). Among patients with low MUST scores determined during the index visit, identification of ≥2 IBD-MR factors was strongly associated with malnutrition development (OR = 8.65, 95%CI 2.21–33.82, *p* = 0.002). (4) Conclusions: We identified IBD-related risk factors for malnutrition, highlighting the need for a disease-specific malnutrition screening tool, which may increase malnutrition risk detection.

## 1. Introduction

The prevalence of malnutrition in inflammatory bowel diseases (IBD) is estimated to be between 6.1% and 69.7% depending on the definition used, the type of IBD, the clinical setting and disease activity [1,2]. Malnutrition and sarcopenia are associated with poor clinical outcomes, hospital admissions, response to therapy and quality of life [1,3,4]. Among hospitalized patients, malnutrition is an independent risk factor for venous thromboembolism, non-elective surgery, longer admission and increased mortality [5]. Due to its high prevalence and associated risks, early detection of IBD patients at risk of malnutrition development is of high importance. Malnutrition is a target for primary and secondary prevention, and patients with IBD are advised to be screened for malnutrition at diagnosis and annually [5]. The most common screening tool for malnutrition risk, and one of the most common tools used in practice, is the Malnutrition Universal Screening Tool (MUST), incorporating body mass index (BMI), recent weight loss and inadequate nutritional intake [6]. Other screening tools, such as the Malnutrition Screening Tool (MST) used for hospitalized patients [7], the Mini Nutritional Assessment—Short Form (MNA-SF) for geriatric patients [8], and the Nutrition Risk Screening (NRS-2002) for assessment of severely ill patients [9], all use relatively similar parameters and provide a numerical score to categorize risk of malnutrition (Appendix A). Additional etiologies for impaired nutritional status among IBD patients include reduced intake due to disease symptoms, malabsorption, enteric nutrient loss, increased basal energy expenditure and certain medications [10]. Further, a patient’s avoidance of eating may be due to fasting during medical procedures, and from prolonged restrictive diets [11]. Therefore, malnutrition and sarcopenia are also prevalent among IBD patients in clinical remission [2]. These IBD-specific malnutrition risk factors are not incorporated into any of the current malnutrition screening tools. Therefore, we aimed to identify IBD-related risk factors of malnutrition development, in the general IBD clinic setting.

## 2. Methods

### 2.1. Study Design

A retrospective matched case-control study. All data were collected from medical files. The study was approved by the local ethics committee of the Tel Aviv Medical Center (TLVMC).

### 2.2. Study Population

The study population included IBD patients treated at the IBD unit of the Department of Gastroenterology and Hepatology of the TLVMC in Israel. During their routine clinical visits, patients are thoroughly evaluated for their clinical disease manifestations and their nutritional status. Weight, weight change, nutritional intake and the MUST score are documented. Both cases and controls were identified using a systematic search protocol of all IBD patients >18 years of age, arriving at the clinic for ≥2 consecutive physician consultations per year, between 2017 and 2020. Cases and controls had normal nutritional status during their first visits to the clinic (index visit), defined as 18.5 kg/m^2^ ≤ BMI ≤ 27 kg/m^2^, and did not answer malnutrition criteria of weight loss (as follows). At the following consecutive clinic visit (diagnostic visit), control participants maintained normal nutritional status, whereas cases developed malnutrition, defined according to the European Society for Clinical Nutrition and Metabolism (*ESPEN*) criteria [12] as one of the following: (1) BMI < 18.5 kg/m^2^; (2) a combination of weight loss >5% during 3 months or weight loss >10% indefinite of time with BMI < 20 kg/m^2^ for patients <70 years or BMI < 22 kg/m^2^ for patients >70 years [5] (Figure 1).

Exclusion criteria for both cases and controls included: lack of information regarding medical or nutritional status evaluation in either consecutive clinical visit; patients with a stoma, pregnancy or malnutrition diagnosed during the index visit. Controls were also excluded if they had developed malnutrition within 2 months post-diagnostic visit.

Controls and cases were matched in a 1:1 ratio, based on age (±5 years), gender, disease (CD/UC/Pouch) and time of diagnostic clinic visit.

### 2.3. Data Collection

Detailed information was gathered from medical files (physician, nurse, dietician visits and multi-disciplinary clinic visits) for both the diagnostic clinic visit and the prior index visit. Information included details of demographics; disease phenotype at visit; and medical history—including chronic general diseases (diabetes mellitus, heart disease, renal disease or liver disease), chronic gastrointestinal (GI) disease (celiac, food intolerance, irritable bowel syndrome or gastric reflux disease) and psychological comorbidities (depressive disorder, anxiety disorder or use of any psychiatric medication); surgical history; disease characteristics; endoscopic results (documented as simple endoscopic score (SES-CD) for CD, MAYO score for UC and pouch disease activity index (PDAI) for pouchitis); ±12 months from the index visit; physician global assessment (PGA); clinical disease activity; disease symptoms and manifestations (during the index visit); biochemical disease activity and blood test results (within 6 months of clinic visit); and medical treatment (at visit). Nutritional intake related symptoms (dental problems, oral aphtha, vomiting, constipation, nausea, dysphagia, diarrhea, pain on eating, loss of appetite and early fullness on eating); annual healthcare utility (number of physician/multidisciplinary clinics used—such as IBD pregnancy, IBD dermatology, IBD rheumatology, IBD surgery—per year, prior to the index visit); reported nutritional intake from diet and supplementation, weight and weight loss; and the MUST score, were documented. Inadequate nutritional intake was determined if decreased or restricted eating was documented for any reason by any healthcare giver.

The etiology for inadequate nutritional intake was categorized as one of the following: dental problems, oral aphtha, vomiting, constipation, nausea, dysphagia, diarrhea, pain on eating, loss of appetite, early satiety/fullness. If ≥2 etiologies were documented, the patient was categorized as suffering from multiple etiologies.

Continuous variables were categorized into dichotomous variables according to the study sample median: BMI at baseline (BMI ≤ 22 kg/m^2^, generating two categories: 18.5 ≤ BMI ≤ 22 kg/m^2^ and BMI > 22 kg/m^2^) and annual healthcare utility (≥5 or <5 physician/multidisciplinary clinic visits per year).

### 2.4. Statistical Analysis

Sample size was calculated as 118 patients using the WINPEPI program assuming-α = 5% and 1 − β = 80%, and according to previous reports of malnutrition proportions among IBD patients with and without an active disease [13,14,15].

All statistical analyses were performed using SPSS version 27.0 for Windows (SPSS Inc., Chicago, IL, USA). Continuous variables are presented as means ± SDs, and nominal variables as proportions.

Serum hemoglobin (HB) level; percentage of weight change; and the differences in BMI, HB, white blood cell (WBC) count and simple colitis clinical activity index (SCCAI) between the two visits were distributed normally. All the other continuous variables were not distributed normally according to Kolmogorov–Smirnov normality test. Cases and controls were compared during the index visit using univariate analysis: the independent samples *t*-test or Mann–Whitney U tests for parametric and non-parametric continuous variables, respectively, and the Chi-square test for categorized data, were used to compare demographic and disease-related characteristics between cases and controls.

Multivariate logistic regression analysis was used to identify independent IBD-related risk factors for malnutrition development, after adjustments for potential confounders and the MUST score. Parameters which were examined by multivariate analysis were those which significantly differed between cases and controls in univariate analysis. These included steroid therapy, elevated CRP, elevated calprotectin, following an elimination diet, high annual healthcare utility, endoscopic disease activity, high stool frequency, moderate–severe abdominal pain and 18.5 ≤ BMI ≤ 22 kg/m^2^. HBI and PGA scores differed between the groups in univariate analysis but were not included in multivariate analysis, since HBI is CD specific, and the PGA may be physician subjective. All parameters which were significantly associated with malnutrition development in multivariate analysis were entered together into a single model with automated parameter elimination. In the final multivariate logistic regression model, risk factors for malnutrition development were adjusted for potential confounders and for one another. Of all significantly associated risk factors which represent disease activity (endoscopic disease activity, high CRP levels, steroid therapy), endoscopic disease activity was chosen to be included as the most IBD specific parameter of disease activity.

The number of IBD-related malnutrition risk factors was summed, and the malnutrition development risk (IBD-MR score) was accordingly categorized as: low risk (0 factors), medium risk (1 factor) and high risk (2 or 3 factors).

## 3. Results

Of the 1596 patients who were treated at the IBD unit of the TLVMC between the years 2017 and 2020, 130 patients were defined as having malnutrition. Of these, 59 patients met the inclusion criteria as cases and had normal nutritional status during the index visit. Matched control patients (*n* = 59) were selected from the remaining 1466 patients who had normal nutritional status throughout the two consecutive clinic visits, and who were individually matched to cases. A flowchart of study population selection incorporating a timeline of physician consultation visits, from which data were collected, is depicted in Figure 1.

### 3.1. Baseline Characteristics of the Study Population and a Comparison between Cases and Controls

Cases and controls did not differ in demographics, disease duration, disease phenotype or medical history (Table 1, Appendix A). During the index visit, cases were characterized by high annual healthcare utility (≥5 physician/multidisciplinary clinic visits per year), and higher proportions of steroid treatment, endoscopic, biochemical and clinical disease activity (Table 1).

During the index visit, both cases and controls had normal mean BMI (range 18.5–27.40 kg/m^2^), and levels of serum albumin and blood hemoglobin. Nonetheless, cases had a significantly lower mean BMI, and had higher proportions of reported weight loss during the 6 months prior to the index visit.

Compared to controls, higher proportions of cases were referred to a dietician, and were classified as having medium–high risk of malnutrition development according to MUST score. Cases had higher proportions of multiple etiologies for inadequate nutritional intake, classification of inadequate nutritional intake, elimination diet followed and regular use of enteral nutritional supplementation (Table 2).

### 3.2. Clinical, Laboratory and Nutritional Parameters throughout Follow-Up among Cases and Controls

The mean time interval between the visits was 6.2 ± 3.0 months, during which cases lost 5.3 ± 2.3 kg of body weight and controls gained 0.2 ± 2.3 kg (*p* < 0.001). Changes in mean hemoglobin level, clinical and biochemical disease activity markers differed significantly between cases and controls during the follow-up period, whereas other nutritional and clinical characteristics did not differ between visits in both cases and controls (Table 3).

### 3.3. Multivariate Analysis of Risk Factors for Malnutrition Development

Multivariate analysis detected significant associations between patient and disease characteristics during the index visit, and malnutrition development during the follow-up period. Endoscopic disease activity (OR = 7.30, 95%CI 1.80–28.14), high annual healthcare utility (OR = 5.58, 95%CI 1.79–17.40), abdominal pain (OR = 3.41, 95%CI 1.03–11.27), 18.5 ≤ BMI ≤ 22 kg/m^2^ (OR = 9.53, 95%CI 3.20–28.37), CRP > 0.5 (OR = 9.15, 95%CI 2.39–35.05) and steroid therapy (OR = 7.23, 95%CI 1.75–29.84) were all positively associated with malnutrition development after adjustment for disease duration, age of diagnosis and MUST score (Appendix A).

We used a final multivariate logistic regression model to incorporate all significantly and independently associated risk factors for malnutrition among IBD patients. These included endoscopic disease activity, high annual healthcare utility, BMI ≤ 22 kg/m^2^ and the number of these risk factors (Table 4).

When assessing these risk factors among a subpopulation of patients with a MUST score of 0 during the index visit, the number of IBD-MR factors was positively associated with malnutrition development, after adjustment for disease duration and age of diagnosis (OR = 3.82, 95%CI 1.60–9.08, *p* = 0.002). Identification of ≥2 IBD-MR factors during the index visit was strongly associated with the odds of malnutrition development (OR = 8.65, 95%CI 2.21–33.82, *p* = 0.002).

## 4. Discussion

Malnutrition is a common complication of IBD, potentially impacting disease morbidity, response to therapy and mortality [14,17,18,19]. Clinical recommendations for nutritional therapy for IBD patients include assessment of nutritional status, and malnutrition risk screening upon diagnosis of IBD and annually [5]. However, to date, there are no published recommendations for use of disease-specific malnutrition risk screening tools [20]. In this study, IBD patients who were of normal nutritional status and developed malnutrition were compared to age, gender and disease-matched controls who maintained normal nutritional status. A 18.5 ≤ BMI ≤ 22 kg/m^2^, moderate–severe endoscopic disease activity, high annual healthcare utility and the sum of these three parameters, were positively associated with malnutrition development, independently of each other and of MUST score.

Inadequate dietary intake, following an elimination diet, enteral nutrition supplementation and weight loss were also associated with malnutrition development. These parameters are incorporated in common malnutrition risk screening tools, such as the NRS-2002, MUST, MST and NRI (Appendix A). On the other hand, clinical characteristics of disease activity, such as abdominal pain and stool frequency, which were positively associated with malnutrition development, are only partly measured by the SaksIBD, but not by other screening tools (Appendix A). Additional parameters that were associated with malnutrition development and are not incorporated into any of the existing malnutrition risk screening tools include physician global assessment, steroid therapy, endoscopic disease activity and annual healthcare utility. Endoscopic disease activity, the target of treatment in IBD [21], may contribute to the development of malnutrition by various mechanisms, such as malabsorption, enteric nutrient loss, reduced energy intake due to disease manifestations [22] and sarcopenia [23,24].

Adjusted for each other and for the patient’s baseline MUST score, a 18.5 ≤ BMI ≤ 22 kg/m^2^, high annual healthcare utility, endoscopic disease activity and the IBD-MR, which represents the accumulation of these three parameters, were positively associated with malnutrition development. The associations were strong and significant within the entire cohort population, and even more so among a subpopulation of patients which had the MUST score of 0 during the index visit. This implies that current malnutrition risk screening tools, which do not take into account important IBD-related characteristics, cannot fully categorize malnutrition risk in IBD patients. Previous reports have shown limited MUST performances in IBD, including misclassification of up to one third of patients who eventually developed malnutrition as being at low risk of developing malnutrition [25,26,27]. These studies suggest that the MUST might misclassify some IBD patients who eventually develop malnutrition, lowering the odds of referral to prevention therapy by healthcare givers. This is emphasized by a relatively low 20% referral rate to dietary therapy among cases in our results.

To our knowledge, only two IBD-malnutrition risk scores have been published. The MIRT, which includes BMI, weight loss and CRP, has been validated against the subjective global assessment (SGA), which is used to assess current malnutrition and not to predict future risk of malnutrition [28]. The SaksIBD, which incorporates weight loss, food restrictions and disease symptoms, has been validated against the MUST, and against retrospective dieticians’ and physicians’ subjective judgments of nutritional status [25]. To our knowledge, our study was the first to assess malnutrition risk factors based on a direct comparison between patients who developed malnutrition and those who maintained normal nutritional status, without the use of a malnutrition risk proxy, enabling the predictive criteria’s validity. Interestingly, during the index visit, all patients had normal BMI and biochemical measures of nutritional status, and though 15% of patients had previously lost > 5% of their body weight, none were considered malnourished according to the ESPEN criteria [5].

Importantly, the associations shown in this study might be underestimations of the true associations with malnutrition development, as cases showed higher rates of malnutrition preventive interventions, such as referral to a dietician, use of partial enteral nutrition and therapy upgrading. As seen, these measures were not able to prevent malnutrition development among cases.

The limitations of this study include the potential information bias of a case-control study. This bias was probably small due to a meticulous and standardized data collection protocol, with systematic data validation, so potential data handling bias was minimized. Data were extracted from all hospital visits—including visits with nurses and dieticians, and visits to the emergency room and other supporting clinics of the IBD unit—so that the missing data bias would be minimal. Data were gathered by one observer, in the same manner in consecutive cases and controls, to prevent differential information bias. Still, differential misclassification of dietary measures among cases is possible and could not be ruled out. A non-differential misclassification bias of nutritional status may have existed, since it was determined based on BMI and weight loss alone, without information regarding body composition, which has recently been at the center of nutritional evaluation recommendations, such as the Global Leadership Initiative on Malnutrition (GLIM) Criteria [29]. Additionally, an important limitation of this study may be the low availability of data regarding endoscopic and biochemical disease activity, and medical information which might be overlooked in clinical practice, such as mental disorders, which might limit our method’s application as a malnutrition screening measure.

On the other hand, strengths of our study included recruitment of cases and controls from the same population, with comparable demographic and medical history characteristics, thereby minimizing potential selection bias and residual confounding. In terms of external validity, this study population was selected from the adult population of patients treated in a tertiary hospital; thus, our findings cannot be generalized to the entire IBD population, due to potential referral filter bias. However, the IBD-related characteristics which were found here to be strongly and independently associated with malnutrition development are universal, and not necessarily associated with the tertiary nature of our clinic. Parameters of the IBD-MR score reflect disease activity and management, and may therefore be relevant for all IBD patients. Additional advantages included our extensive and meticulous data collection, and the direct measuring of malnutrition as the study outcome, enabling predictive criterion validity for the IBD-MR score.

In conclusion, our study elaborated on IBD-related risk factors for malnutrition, highlighting the need for a disease-specific screening tool, which takes into account the unique clinical manifestations, dietary habits and chronic inflammatory nature of this population. These findings should be further validated in a larger, prospective study, which might enable formulating a prediction model for malnutrition among IBD patients.

## Figures and Tables

**Figure 1 nutrients-13-04098-f001:**
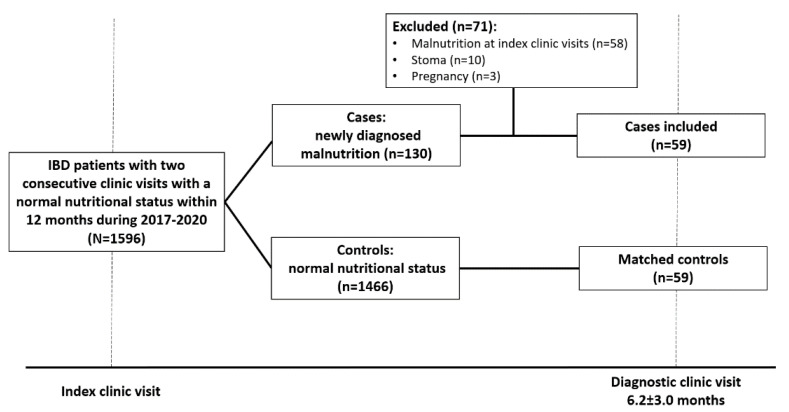
Data collection timeline and flowchart of study population selection.

**Table 1 nutrients-13-04098-t001:** Demographic and clinical characteristics of the study population during the index visit.

	Total Population *n* = 118	Controls *n* = 59	Cases *n* = 59	*p*
**Demographic characteristics**				
Age (years) (mean ± std)	38.7 ± 17.3	38.5 ± 17.2	38.9 ± 17.6	0.966
Gender (% women)	66.1	66.1	66.1	1.000
Current smoking (%)	16.9	13.6	20.3	0.326
**Disease characteristics (%)**				
CD	64.4	64.4	64.4	1.000
UC	23.7	23.7	23.7	
Pouch	11.9	11.9	11.9	
Disease duration (years) (mean ± std)	11.2 ± 9.4	10.6 ± 8.6	11.7 ± 10.3	0.668
Past intestinal resection (%)	27.1	27.1	27.1	1.00
Extra-intestinal manifestations (%)	35.6	39.0	45.8	0.660
**Medical treatment and background (%)**				
5ASA	28.0	27.1	28.8	0.837
Immunomodulators	16.1	20.3	11.9	0.210
Advanced therapy (biologics and small molecules)	59.3	67.8	50.8	0.061
**Steroid therapy**	**15.3**	**6.8**	**23.7**	**0.001**
Chronic diseases ^a^ (%)	22	16.9	27.1	0.183
Chronic GI disease ^b^ (%)	19.5	18.6	20.3	0.763
Psychological comorbidities ^c^ (%)	15.3	13.6	16.9	0.609
**Disease activity at the index visit (mean ± std)**				
**Harvey Bradshaw index (HBI) (*n* =** **76)**	**5.1 ± 4.3**	**3.9 ± 3.5**	**6.1 ± 5.0**	**0.035**
Simple clinical colitis activity index (SCCAI) (*n* = 25)	4.2 ± 3.9	2.8 ± 3.0	5.9 ± 4.3	0.092
Clinical Pouch Disease Activity Index (cPDAI) (*n* = 14)	1.7 ± 1.8	1.0 ± 1.4	2.4 ± 1.9	0.161
**CRP (mg/dl) (*n* = 100)**	**1.3 ± 1.8**	**0.8 ± 1.3**	**1.8 ± 2.1**	**<0.001**
**Calprotectin (µg/gr) (*n* = 40)**	**353.7 ± 524.5**	**284.3 ± 592.9**	**469.3 ± 374.6**	**0.008**
**WBC (µL/3*10) (*n* = 106)**	**7.8 ± 2.5**	**7.3 ± 2.5**	**8.2 ± 2.4**	**0.033**
**High annual healthcare utility ^d^ (%)**	**27.1**	**15.3**	**39**	**0.004**
**Endoscopic disease activity at the index visit (*n* = 79) (%)**				
**Inactive disease**	**21.5**	**39.5**	**4.9**	**<0.001**
**Mild disease**	**35.4**	**36.8**	**34.1**	
**Moderate disease**	**32.9**	**21.1**	**43.9**	
**Severe disease**	**10.1**	**2.6**	**17.1**	

^a^ Chronic disease was defined as at least one of the following: type 2 diabetes millets, heart disease, renal disease or liver disease. ^b^ Chronic GI disease was defined as at least one of the following: celiac, food intolerance, irritable bowel syndrome or gastric reflux disease. ^c^ Psychological comorbidities were defined as at least one of the following: depressive disorder, anxiety disorder or use of any psychiatric medication. ^d^ High annual healthcare utility was defined as ≥5 physician/multidisciplinary clinic visits per year. Abbreviations: CD—Crohn’s disease, UC—ulcerative colitis, GI—gastrointestinal, CRP—C reactive protein, WBC—white blood cells, HBI—Harvey Bradshaw index, SCCAI—simple clinical colitis activity index, cPDAI—clinical pouch disease activity index.

**Table 2 nutrients-13-04098-t002:** Nutritional status and intake characteristics during the index visit.

	Total Population (*n* = 118)	Controls (*n* = 59)	Cases (*n* = 59)	*p*
**Nutritional status characteristics (mean ± std)**				
Hemoglobin (mg/dL) (*n* = 113)	12.9 ± 1.2	13.1 ± 1.2	12.6 ± 1.2	0.06
Serum Albumin (mg/dL) (*n* = 84)	4.0 ± 0.4	4.0 ± 0.4	3.9 ± 0.4	0.114
**BMI (kg/m^2^)**	21.8 ± 2.2	23.2 ± 1.8	20.3 ± 1.5	<0.001
**Weight loss during 6 months prior to the index visit (%)**				
**≤5% weight loss**	85.6	93.2	78.0	0.035
**5–10% weight loss ^a^**	11.0	6.8	15.3	
**≥10% weight loss ^b^**	3.4	0.0	6.8	
**Physician referral to dietician**	13.6	6.8	20.3	0.031
**Inadequate nutritional intake**	15.3	1.7	28.8	<0.001
**Multiple etiologies for inadequate nutritional intake ^c^**	15.3	6.8	23.7	0.010
**Enteral nutrition supplementation**	7.6	1.7	13.6	0.032
Iron supplementation	15.3	15.3	15.3	1.000
**Follow elimination diet**	29.7	20.3	39.0	0.027
**MUST score at the index visit ^d^ (%)**				
**Low risk**	63.6	93.2	33.9	<0.001
**Medium risk**	31.4	6.8	55.9	
**High risk**	5.0	0.0	10.2	

^a^ Patients lost 5–10% of their weight throughout a period longer than 3 months (*n* = 13). None met the definition of malnutrition. ^b^ Patients lost more than 10% of their body weight, but maintained a BMI above 20 kg/m^2^ (*n* = 4). **^c^** Multiple etiologies for inadequate nutritional intake were determined with documentation of ≥2 of the following: dental problems, oral aphtha, vomiting, constipation, nausea, dysphagia, diarrhea, pain on eating, loss of appetite, early fullness on eating. ^d^ MUST score was determined as previously described [6,16], incorporating BMI, unintentional weight loss over 3 months and inadequate eating anticipated for 5 days. Abbreviations: BMI—body mass index, MUST—malnutrition universal screening tool.

**Table 3 nutrients-13-04098-t003:** Changes (Δ) in nutritional and clinical characteristics between visits, among cases and controls.

	Controls *n* = 59	Cases *n* = 59	*p*
**Clinical characteristics (mean ± std)**			
**BMI (kg/m^2^) (*n* = 118)**	0.2 ± 1.0	−1.9 ± 0.8	<0.001
**Percent of weight change (% of body weight at index visit) (*n* = 118)**	0.3 ± 3.6	−9.3 ± 3.8	<0.001
**Hemoglobin (mg/dL) (*n* = 107)**	0.0 ± 0.9	−0.4 ± 1.2	0.016
Serum Albumin (mg/dL) (*n* = 62)	−0.0 ± 0.2	−0.1 ± 0.4	0.076
Pain VAS score (*n* = 96)	0.4 ± 2.8	0.7 ± 2.9	0.817
Stool frequency (number/day) (*n* = 111)	0.8 ± 3.6	0.76 ± 6.7	0.292
**Harvey Bradshaw index (HBI) (*n* = 75)**	1.0 ± 5.8	3.7 ± 6.7	0.018
Simple clinical colitis activity index (SCCAI) (*n* = 25)	0.2 ± 3.5	1.9 ± 2.8	0.225
Clinical Pouch Disease Activity Index (cPDAI) (*n* = 14)	0.8 ± 1.0	−0.2 ± 0.9	0.073
**CRP (mg/dL) (*n* = 88)**	−0.4 ± 4.8	1.8 ± 3.9	<0.001
Calprotectin (µg/gr) (*n* = 25)	−62.7 ± 852.2	192.9 ± 483.7	0.189
WBC (µL/3*10) (*n* = 100)	−0.3 ± 2.0	0.2 ± 2.1	0.165

Changes in all parameters are calculated as the difference of each parameter between index and diagnostic visits (difference = diagnostic − index). Abbreviations: BMI—body mass index, VAS—visual analogue scale, CRP—C reactive protein, WBC—white blood cells, HBI—Harvey Bradshaw index, SCCAI—simple clinical colitis activity index, cPDAI—clinical pouch disease activity index.

**Table 4 nutrients-13-04098-t004:** Adjusted associations between IBD characteristics and malnutrition development (*n* = 79).

	OR (95%CI) *p*
18.5 ≤ BMI ≤ 22 kg/m^2^	4.71 (1.13–19.54) 0.032
High annual healthcare utility ^a^	5.67 (1.02–31.30) 0.047
Endoscopic disease activity ^b^	5.49 (1.28–23.56) 0.022
Number of IBD-related malnutrition risk factors (IBD-MR score) ^c^	7.39 (2.60–20.94) <0.001

ORs were adjusted for disease duration, age of diagnosis, MUST score at index visit, and for one another. ^a^ High annual healthcare utility was defined as ≥5 physician/multidisciplinary clinic visits per year. ^b^ Endoscopic disease activity was defined as moderate/severe endoscopic disease. ^c^ The number of IBD-related malnutrition risk factors was calculated as the sum of identified factors: 18.5 ≤ BMI ≤ 22 kg/m^2^, endoscopic disease activity, high annual healthcare utility. ORs were adjusted for disease duration, age of diagnosis and MUST score at index visit. Abbreviations: BMI—body mass index, MUST—malnutrition universal screening tool, IBD-MR—IBD-related malnutrition risk factors.

## Data Availability

The data presented in this study are available on request from the corresponding author. The data are not publicly available due to ongoing analysis for further publication.

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
