# Peer review of "Risk Factors for Malnutrition among IBD Patients"

_nutrients, 2021, doi:10.3390/nu13114098_

Round 1

Reviewer 1 Report

Malnutrition is a prevalent condition in IBD. The authors perform a careful clinical study to assess risk factors for malnutrition in a cohort of IBD patients. The paper is well written and the results are clearly presented. However, while the data are interesting, some over-interpretation of the data needs to be corrected, especially since no validation cohort has been recruited. Further, handling of missing values needs to be clearly described. 

Major comments:
1. If only a training set is available, fitting of a predictive model is not appropriate (there is the risk of overfitting). Therefore, the numbers regarding AUC, sensitivity and specificity of their new IBD-MR score are meaningless at best (if not to say potentially misleading) and should not be reported (multivariable models and odds ratios are fine of course). Especially, comparison with the MUST score is misleading here (since this score has not been trained for this cohort). If the authors want to develop a predictive model, the data should be divided into a training and a validation set (which might be difficult with the number of patients available). An independent validation cohort would be even better. 
2. Missing data. Missing data in my experience are a nearly intractable problem in a retrospective study but this seems to be insufficiently addressed. The authors should either state in the methods that for all measures data from all patients were available; if not, the number of missing values should be stated for every measure in every table.
3. On a similar note: Endoscopy scores/ clinical sores: what was the time difference between index visit and the last endoscopy? Did any thresholds for the timing apply (e.g < 3 months)?
4. If indeed the number of missing values (also for calprotectin, pain-VAS, HBI, diarrhea) were negligible, the authors might have incorporated a structured approach to clinical care with standardized values acquired for every patient at every visit and it would be interesting to read about this. 
5. Depression/ anxiety/ PTSD are highly prevalent in IBD patients. However, they are infrequently addressed in clinical practice and cases might have been missed if no structured approach (e.g. a questionnaire) was used (this is a potential limitation of this study).  
6. Multivariable analysis: of the acquired data only a limited set of potential predictive variables were considered. Further, it is not clear, why some parameters (I.e. HBI, psychiatric co-morbidity) were not considered. Another option would be to include all potential predictors into a single model and use automated parameter elimination.  

Author Response

25.10.21

Dear editor of Nutrients,

We would like to thank the editors and reviewers for the beneficial comments. We appreciate the opportunity to submit a revised version of the manuscript ID nutrients-1398050, entitled “Risk factors for malnutrition among IBD patients”.

We have made corrections in accordance to the reviewers' comments. All the corrections in the manuscript are highlighted in yellow. Line numbers were added to the manuscript for the reviewer's convenience.  

Attached is a point-by point reply to the reviewers.

Comments and Suggestions for Authors

Malnutrition is a prevalent condition in IBD. The authors perform a careful clinical study to assess risk factors for malnutrition in a cohort of IBD patients. The paper is well written and the results are clearly presented. However, while the data are interesting, some over-interpretation of the data needs to be corrected, especially since no validation cohort has been recruited. Further, handling of missing values needs to be clearly described.

Major comments:

Comment 1. If only a training set is available, fitting of a predictive model is not appropriate (there is the risk of overfitting). Therefore, the numbers regarding AUC, sensitivity and specificity of their new IBD-MR score are meaningless at best (if not to say potentially misleading) and should not be reported (multivariable models and odds ratios are fine of course). Especially, comparison with the MUST score is misleading here (since this score has not been trained for this cohort). If the authors want to develop a predictive model, the data should be divided into a training and a validation set (which might be difficult with the number of patients available). An independent validation cohort would be even better.

Answer: We thank the reviewer for this important insight. We have deleted all comparisons between the IBD-MR and the MUST score (abstract P1, statistical methods P4, all of section 3.4 in the results P9). We added a comment regarding a need to further validate our results in future larger prospective cohorts, which could enable formulating a prediction model for malnutrition development among IBD patients: “These findings should be further validated in a larger, prospective study, which might enable formulating a prediction model for malnutrition among IBD patients” (page 11).

To better emphasize the clinical relevance of the IBD-MR and the MUST score, we added results from a multivariate analysis aimed to associate the IBD-MR and malnutrition development among patients with a low MUST score (results page 9, discussion page 10).

Comment 2. Missing data. Missing data in my experience are a nearly intractable problem in a retrospective study but this seems to be insufficiently addressed. The authors should either state in the methods that for all measures data from all patients were available; if not, the number of missing values should be stated for every measure in every table.

Answer: Thank you for this important comment. As can be expected in a retrospective study, not all variables were available for all patients. Demographic information, disease characteristics, medical treatment, BMI, weight loss, the MUST score, and annual health-care utility were documented for all patients. Unfortunately, biochemical measures and endoscopic examinations were not available for all study patients. We have added the number of available values for every measure in every table, and emphasized this limitation of the study in the discussion: “Also, an important limitation of this study may be the low availability of data regarding endoscopic and biochemical disease activity, or medical information which might be overlooked in clinical practice such as mental disorders, which might limit its application as a malnutrition screening measure” (page 11).

Comment 3. On a similar note: Endoscopy scores/ clinical sores: what was the time difference between index visit and the last endoscopy? Did any thresholds for the timing apply (e.g < 3 months)?

Answer: Clinical disease activity was documented at the index visit. Endoscopic disease activity was documented ±12 months from the index visit, thought 86% of participants with a documented endoscopic examination had undergone the procedure within ±6 months from the index visit. Clarifications were made in the methods: “endoscopic results (documented as simple endoscopic score (SES-CD) for CD, MAYO score for UC, and pouch disease activity index (PDAI) for pouchitis, ±12 months from the index visit, Physician Global Assessment (PGA), and clinical disease activity, disease symptoms and manifestations (at the index visit), biochemical disease activity and blood test results (within 6 months of clinic visit) and medical treatment (at visit)” (page 3).

Comment 4. If indeed the number of missing values (also for calprotectin, pain-VAS, HBI, diarrhea) were negligible, the authors might have incorporated a structured approach to clinical care with standardized values acquired for every patient at every visit and it would be interesting to read about this.

Answer: Biochemical measures and endoscopic examinations were not available for all study patients. We have added the number of available values for every measure in every table, and emphasized this limitation of the study in the discussion. No imputation approach was used for missing data.

Comment 5. Depression/ anxiety/ PTSD are highly prevalent in IBD patients. However, they are infrequently addressed in clinical practice and cases might have been missed if no structured approach (e.g. a questionnaire) was used (this is a potential limitation of this study). 

Answer: Thank you for this important comment. We agree that the main limitation of the study is its retrospective nature, and lack of structured data collection method for some of the information gathered. We assume that since data was extracted from all hospital visits including nurse, dietician, emergency room and other supporting clinics of the IBD unit, missing data bias would be minimal, especially regarding diagnosed mental conditions, which as mentioned, are common and should not be overlooked among IBD patients. We have emphasized this potential information bias in the discussion: “an important limitation of this study may be the low availability of data regarding endoscopic and biochemical disease activity, or medical information which might be overlooked in clinical practice such as mental disorders, which might limit its application as a malnutrition screening measure” (page 11).  

Comment 6. Multivariable analysis: of the acquired data only a limited set of potential predictive variables were considered. Further, it is not clear, why some parameters (I.e. HBI, psychiatric co-morbidity) were not considered. Another option would be to include all potential predictors into a single model and use automated parameter elimination.

Answer: We thank the reviewer for this important comment. We have added a clarification of the statistical analysis performed by multivariate analysis in section 2.4: “Parameters which were examined by multivariate analysis were those which significantly differed between cases and controls in univariate analysis. These included steroid therapy, elevated CRP level, elevated Calprotectin level, following an elimination diet, high annual health-care utility, endoscopic disease activity, high stool frequency, moderate-severe abdominal pain, and 18.5<BMI≤22 kg/m2. HBI and PGA scores differed between the groups in univariate analysis but were not included in multivariate analysis since HBI is CD specific, and the PGA may be physician subjective. All parameters which were significantly associated with malnutrition development in multivariate analysis were entered together into a single model with automated parameter elimination” (page 3).   

Sincerely,

Naomi Fliss Isakov, RD PhD

IBD unit, department of Gastroenterology and Hepatology

Tel Aviv Medical Center

Tel Aviv University

email: naomifl@tlvmc.gov.il

Tel.: +97-(23)-6947305

Address: Weizmann 6, Tel Aviv, Israel

Prof. Nitsan Maharshak, MD

Head of the IBD unit, department of Gastroenterology and Hepatology

Tel Aviv Medical Center

Tel Aviv University

email: nitsanm@tlvmc.gov.il

Tel.: +97-(23)-6947305

Address: Weizmann 6, Tel Aviv, Israel

Reviewer 2 Report

The paper "Risk factors for malnutrition among IBD patients" is a retrospective case-control study based on the analysis of medical files of adult IBD patients treated in a tertiary hospital that aimed to identify disease-related factors for malnutrition. The authors managed to calculate the new IBD malnutrition risk score (IBD-MR) that turned out to be more sensitive that the MUST score used for this reason until now. As no IBD-specific malnutrition risk screening tool was available to date, the IBD-MR score was long-anticipated and seems to be promising, though its utility has to be confirmed in large-scale samples.

The paper is well-written and appropriately constructed. The results are clearly presented in the main text and graphically in 2 figures and 3 tables and sufficiently discussed.  

The abbreviation SCCAI when used for the first time in the text (page 3) should be explained to the readers. I have no other objections regarding the study.

Author Response

Dear editor of Nutrients,

We would like to thank the editors and reviewers for the beneficial comments. We appreciate the opportunity to submit a revised version of the manuscript ID nutrients-1398050, entitled “Risk factors for malnutrition among IBD patients”.

We have made corrections in accordance to the reviewers' comments. All the corrections in the manuscript are highlighted in yellow. Line numbers were added to the manuscript for the reviewer's convenience.  

Attached is a point-by point reply to the reviewers.

Editor’s comment: Please add Keywords in your manuscript.

Answer: Key words have been added to the end of the abstract (Page 1).

Reviewer 1

Comments and Suggestions for Authors

The paper "Risk factors for malnutrition among IBD patients" is a retrospective case-control study based on the analysis of medical files of adult IBD patients treated in a tertiary hospital that aimed to identify disease-related factors for malnutrition. The authors managed to calculate the new IBD malnutrition risk score (IBD-MR) that turned out to be more sensitive that the MUST score used for this reason until now. As no IBD-specific malnutrition risk screening tool was available to date, the IBD-MR score was long-anticipated and seems to be promising, though its utility has to be confirmed in large-scale samples.

The paper is well-written and appropriately constructed. The results are clearly presented in the main text and graphically in 2 figures and 3 tables and sufficiently discussed. 

Comment: The abbreviation SCCAI when used for the first time in the text (page 3) should be explained to the readers. I have no other objections regarding the study.

Answer: Thank you for this correction, the mistake has been corrected (statistical analysis, page 3).

Sincerely,

Naomi Fliss Isakov, RD PhD

IBD unit, department of Gastroenterology and Hepatology

Tel Aviv Medical Center

Tel Aviv University

email: naomifl@tlvmc.gov.il

Tel.: +97-(23)-6947305

Address: Weizmann 6, Tel Aviv, Israel

Prof. Nitsan Maharshak, MD

Head of the IBD unit, department of Gastroenterology and Hepatology

Tel Aviv Medical Center

Tel Aviv University

email: nitsanm@tlvmc.gov.il

Tel.: +97-(23)-6947305

Address: Weizmann 6, Tel Aviv, Israel

Round 2

Reviewer 1 Report

I agree with the revisions by the authors, I have no further concerns.